# Dalbavancin Boosts the Ability of Neutrophils to Fight Methicillin-Resistant *Staphylococcus aureus*

**DOI:** 10.3390/ijms24032541

**Published:** 2023-01-28

**Authors:** Sara Scutera, Rosaria Sparti, Sara Comini, Francesca Menotti, Tiziana Musso, Anna Maria Cuffini, Valeria Allizond, Giuliana Banche

**Affiliations:** Department of Public Health and Pediatrics, University of Torino, 10125 Turin, Italy

**Keywords:** dalbavancin, methicillin-resistant *Staphylococcus aureus*, polymorphonuclear leukocytes, intracellular killing activity, apoptosis, reactive oxygen species production, cytokine release, neutrophil extracellular traps

## Abstract

Polymorphonuclear leukocytes (PMNs) are the most important cell type involved in the early nonspecific host response to bacterial pathogens. *Staphylococcus aureus* has evolved mechanisms to evade immune responses that contribute to its persistence in PMNs, and acquired resistance to several antimicrobials. Additionally, methicillin-resistant *S. aureus* (MRSA) is one of the most common causes of acute bacterial skin and skin-structure infections (ABSSSIs). Dalbavancin (DBV), a lipoglycopeptide, is indicated for the treatment of ABSSSIs, and has a broad spectrum of action against most microorganisms. Here, we sought to determine the effect of DBV on the neutrophil killing of MRSA and its potential immunomodulating activity. Our results revealed that DBV boosts MRSA killing by acting on both bacteria and PMNs. DBV pre-treatment of PMNs did not change the respiratory burst or degranulation, while an increased trend in neutrophil extracellular traps-associated elastase and in the production of TNFα and CXCL8 was revealed. In parallel, DBV caused a delay in the apoptosis of MRSA-infected neutrophils. In conclusion, we demonstrated a cooperative effect between the antimicrobial properties of PMNs and DBV, thus owing to their immunomodulatory activity. In the choice of the treatment management of serious *S. aureus* infections, DBV should be considered as an outstanding option since it reinforces PMNs pathogen clearance capability by exerting its effect directly, not only on MRSA but also on neutrophils.

## 1. Introduction

Dalbavancin (DBV), a lipoglycopeptide derived from a teicoplanin-like natural antibiotic, interferes with the biosynthesis of the bacterial cell wall by binding to the terminal d-alanyl-d-alanine residues of peptidoglycan precursors with consequent inhibition of both transpeptidase and transglycosylase activities [1,2,3]. In adults, DBV is approved for the treatment of acute bacterial skin and skin-structure infections (ABSSSIs)—a subcategory of complicated skin and soft tissue infections—comprising cellulitis, erysipelas, relevant skin abscesses and wound infections, and that are common in both the community and hospital settings and represent a relevant economic burden to the healthcare systems [4,5,6,7].

The most common pathogens involved in ABSSSIs are Gram-positive bacteria, *Streptococcus pyogenes* and *Staphylococcus aureus*, mainly the methicillin-resistant *S. aureus* (MRSA) strains [6,8,9]. The use of DBV for ABSSSIs treatments provides several advantages over traditional glycopeptides; in particular, the modified chemical structure of DBV results in increased antimicrobial potency (more potent bactericidal activity), also against MRSA, and in a high plasma protein binding with a prolonged elimination half-life and good tissue penetration [1,2,4,5,6,8,10,11].

Polymorphonuclear leukocytes (PMNs) are the most prominent cell type involved in the early, non-specific host response to invading microorganisms [12,13,14,15]. They play a pivotal role in the host defense against staphylococcal infections, exerting a multi-array of microbicide mechanisms including phagocytosis, intracellular killing, antimicrobial molecules and soluble factors (chemokines and cytokines) release [16,17,18,19,20].

Unfortunately, *S. aureus* survival and persistence within PMNs greatly contribute to the pathogenesis and progression of staphylococcal infection [17,21]. *S. aureus* has evolved several strategies to interfere with PMN functions essential for bacterial clearance. Specifically, *S. aureus* has been shown to inhibit neutrophil recruitment, phagocytosis and bacterial killing mediated by reactive oxygen species (ROS), antimicrobial peptides and neutrophil extracellular traps (NETs) production [13,17,22]. Moreover, it appears to alter neutrophil lifespan by promoting or delaying different host-programmed cell death pathways (apoptosis, necrosis, pyroptosis) depending on the stage of infection and virulence determinants [23,24]. Restoring neutrophil antimicrobial potency and avoiding an excessive, harmful inflammatory response, in association with the direct effect of antibiotics on bacteria, could potentially ameliorate the treatment of *S. aureus* infections. In recent years, more studies have been focused on the influence of antibiotics on the function of host immune cells such as monocytes/macrophages and neutrophils [25,26]. For example, macrolides can facilitate the killing of different microorganisms through neutrophil stimulation [27,28]. In fact, an enhanced intracellular killing of MRSA by azithromycin and the azithromycin pro-drug CSY5669 has been recently reported [29]. However, not exhaustive and sometimes contradictory results on the actions of antibiotics in the interaction between *S. aureus* and neutrophils are present in literature [30].

To date, there have been no studies regarding the potential immunomodulating activity of DBV. Firstly, the aim of the present research was to investigate the DBV–MRSA–PMN interactions, and intracellular killing of MRSA was determined in the presence of DBV at different concentrations. Going forward, to distinguish between any separate effects of DBV on MRSA and/or PMNs, we investigated the susceptibility to the killing of DBV pre-treated MRSA and the functional activities of PMN pre-treated with DBV, including MRSA killing activity, viability, ROS production, degranulation, NETs and pro-inflammatory cytokine release.

## 2. Results

### 2.1. Effect of Dalbavancin at Different Concentrations on Neutrophil Intracellular MRSA Killing Activity

Our primary goal was to evaluate the effect of different concentrations of DBV, that are 0.25 µg/mL (MIC) and 0.5 µg/mL (2xMIC), on the intracellular killing activity of human PMNs towards MRSA at increasing timepoints (30, 60 and 90 min), measured as survival index (SI) and as the corresponding killing percentage (Figure 1). In the drug-free controls, PMNs were slightly able to counteract MRSA as demonstrated by a decrease in the intraPMN killing activity with values of 33% (SI = 1.67), 22% (SI = 1.78) and 12% (SI = 1.88), at 30, 60 and 90 min, respectively. The addition of DBV at MIC or 2xMIC significantly (*p* < 0.05 or *p* < 0.001) enhanced the PMN killing capability, reaching values of ~50% (~SI = 1.50) within 90 min of incubation (Figure 1A), in comparison with antibiotic-free controls.

To differentiate between any separate effect of DBV on bacteria or PMNs, in vitro DBV pre-treatment experiments were also performed by exposing each of them to DBV for 60 min, before they were incubated together; thereafter, the intracellular killing activity was assessed. Figure 1B showed that, when MRSA was pre-treated with DBV (at MIC or 2xMIC) before being in contact with PMNs, a boosted killing action was demonstrated starting from 30 min of incubation, with respect to un-pre-treated controls. Whereas, for drug-free controls, a decrease in PMN killing was observed when PMNs were in contact with DBV pre-treated MRSA (at 0.25 µg/mL), in which the killing values significantly (*p* < 0.001) improved to 41% (SI = 1.59), 50% (SI = 1.50) and 50% (SI = 1.50), after 30, 60 and 90 min of incubation. A comparable trend was revealed with 0.5 µg/mL DBV concentration.

Similar results were obtained when PMNs were pre-treated with DBV before assessing its killing activity against MRSA, in which the killing values (*p* < 0.001) saw a significant rise compared to controls reaching 59% (SI = 1.41) after 30 min, 56% (SI = 1.44) after 60 min, and 53% (SI = 1.47) after 90 min of incubations (Figure 1C). Thus, it indirectly demonstrates that DBV penetrates PMNs allowing intra-MRSA killing as well.

After MRSA exposure, the expression of the pro-inflammatory cytokines, specifically TNFα and CXCL8, was evaluated in the supernatants of DBV-pre-treated PMNs, in which an augmented level of release related to both cytokines was recorded, even if it did not reach a statistically significant degree. 

For both cytokines, at the end of the killing assays, an augmented release was measured even if it did not reach a statistically significant difference. Regarding TNFα, a quantification of 27.42 ± 4.90 pg/mL versus 38.52 ± 5.94 pg/mL, for controls and DBV pre-treated-PMNs was obtained, respectively. Whereas, for CXCL8, the quantification was 6037 ± 1710 pg/mL for drug-free controls and 8851 ± 644 pg/mL for DBV pre-treated-PMNs.

### 2.2. Effect of Dalbavancin on Neutrophil Viability

PMNs are characterized by a short lifespan undergoing programmed apoptosis. To verify the effect of DBV on neutrophil apoptosis, resting cells were treated with the antibiotic at MIC (0.25 µg/mL) and at the therapeutically achievable concentration of 25 µg/mL. After 4 h, cell viability was evaluated using the Annexin V FITC/Propidium iodide (PI) staining. As demonstrated in Figure 2A, no differences were observed between untreated and DBV-treated PMNs at both doses. After MRSA phagocytosis, neutrophils can undergo apoptosis to be removed by macrophages for infection resolution [31]. Apoptosis was measured in PMN pre-treated with the higher dose of DBV and then activated with MRSA or PMA. Delayed apoptosis was observed after PMN DBV pre-treatment with a significant decrease in late apoptotic cells and an increase in cell viability. This effect was not observed after treatment with the artificial activator phorbol myristate acetate (PMA) (Figure 2A,B).

### 2.3. Effect of Dalbavancin on ROS Production by Neutrophils

To verify if DBV can modulate ROS production by PMNs, we assessed the respiratory burst by flow cytometry, via DHR123 oxidation. Cells, pre-treated for 60 min with the antibiotic, were left unstimulated or stimulated with PMA or MRSA for 30 min. DBV does not modulate ROS production in resting neutrophils (data not shown). Whereas, the presence of DBV, both at MIC and plasmatic concentration, had a mild effect on ROS, following MRSA and PMA treatment (Figure 3).

### 2.4. Effect of Dalbavancin on NETosis and Degranulation

NETosis is an antimicrobial mechanism representing the release of NETs able to neutralize pathogens [32]. The synthetic molecule PMA is commonly used to stimulate NETosis in vitro, inducing a ROS-dependent suicidal NETosis [33]. The protease neutrophil elastase has been implicated in the formation of NETs and can be used to evaluate NETosis as parallel measurement of released dsDNA.

To understand if DBV might influence NETosis, cells were pre-treated with the drug, at MIC and plasmatic concentration, and then stimulated with PMA to induce NETs formation. An increasing trend in NET-associated elastase, although not significant, was observed in PMNs in the presence of DBV in both resting and activated cells (Figure 4A).

In parallel, neutrophils were evaluated for degranulation, another important feature of PMN activity. Using flow cytometry analysis, the side scatter channel was used as an indicator of the granularity and internal structure of the cells. No difference in the presence of DBV was observed with respect to not stimulated and PMA-treated cells (Figure 4B).

### 2.5. Cytokine Release Pattern of Dalbavancin Pre-Treated PMNs upon MRSA or PMA Stimulation

Finally, we evaluated the effect of the antibiotic on the release of the pro-inflammatory cytokines CXCL8 and TNFα by neutrophils. A significant increase in CXCL8 and TNFα was observed after PMA and MRSA stimulation. Cells pre-treated with DBV at the MIC value showed an increase in cytokines production, both in basal conditions and after stimuli activation. Cells pre-treated with DBV at the plasmatic dose released an amount of cytokines similar to those of not pre-treated cells (Figure 5A,B).

## 3. Discussion

Neutrophils represent the highest number of leukocytes in the human blood of healthy subjects, and they are considered the first line of immune protection after bacteria have penetrated into the epithelial barriers [21,34]. PMNs employ a combination of NADPH oxidase-derived ROS, cytotoxic granule components, antimicrobial peptides and NETs to create a highly dangerous *milieu* that is pivotal for an effective bacterial killing and degradation [30,31,35,36,37]. PMNs cell death and removal by macrophages lead to infection resolution and the avoidance of dangerous inflammation [21,35].

*S. aureus*, a widespread Gram-positive bacterium, is a common commensal microorganism, but it can be responsible for a wide range of infections and inflicts a relevant burden on the healthcare system [30,34,38,39,40]. Despite PMNs being able to eradicate most of the microorganisms, *S. aureus* acquired the ability to escape neutrophil-mediated host defenses by altering neutrophil chemotaxis, phagocytosis and antibacterial activities [21,38,41]. Furthermore, *S. aureus* survival inside PMNs facilitates the dissemination, causing infections distant from the initial site [30]. The *S. aureus* predisposition to develop antibiotic resistance and to escape the human immune effectors, mainly neutrophils, means a search is required for novel therapeutic regimens. In fact, diminished *S. aureus* susceptibility to glycopeptides, especially in MRSA strains, posed a challenge in antimicrobial availability [42]. In this context, dalbavancin, a semisynthetic lipoglycopeptide active against Gram-positive bacteria, including MRSA, has been recently approved for the treatment of severe skin infections [1,2,4,42].

The aim of the present research was to investigate whether dalbavancin could boost neutrophil antibacterial activities, by the in vitro evaluation of intra-MRSA killing activity, PMN viability, ROS production, degranulation and NETs and the pro-inflammatory cytokine release.

We observed that in the drug-free controls, human PMNs harvested from heathy subjects were partially unable to counteract staphylococcal growth, as demonstrated by decreasing killing values (from 33% to 12%) overtime. Notably, our results agree with works reporting that resistant microbial strains, such as MRSA and carbapenem-resistant *Klebsiella pneumoniae*, have an enhanced capacity to circumvent both phagocytosis and killing by human neutrophils [21,40,43,44]. The role of glycopeptides on PMNs functions is debated. In a recently published review, it is reported that glycopeptides seem to not affect PMNs functions [30]; whereas, other studies demonstrate that vancomycin and teicoplanin improve susceptibility to *S. aureus* killing by neutrophils [45,46]. Here, we revealed that DBV is able, at MIC (0.25 µg/mL) and 2xMIC (0.5 µg/mL) concentrations, to enhance intracellular killing of PMNs towards MRSA, reaching values of ~50%. This might be due to a direct DBV alteration of the bacterial external layers, mainly peptidoglycan, which might improve neutrophil phagocytosis. Other literature data confirmed that pre-incubation of *S. aureus* with clindamycin or lincomycin causes an augmented susceptibility to phagocytosis and quicker PMN-induced killing [41].

Several studies discussed the challenge in discerning between the direct intracellular antibacterial effects of antibiotics that enter into PMNs and the effect of antibiotics on antimicrobial neutrophil functions against *S. aureus* [30]. Mandell G.L. and Coleman E. (2001) showed that several antibiotics i.e., azithromycin, ciprofloxacin, but not penicillin G, could penetrate into granulocytes [47]. Similarly, intra-phagocytic bioactivity was shown for clarithromycin [48], as for fosfomycin [49]. Other research demonstrated that glycopeptides show no intracellular bioactivity against *S. aureus* [30,50,51,52]. Here, we performed DBV pre-treatment experiments to discriminate its action on either MRSA or PMNs.

As expected, when MRSA was pre-treated for 60 min with DBV, before the incubation with neutrophils, enhanced killing action was observed with killing percentages ranging from 41% to 50% at both MIC and 2xMIC DBV concentrations, if compared with untreated controls. These data agree well with those of Yamaoka T. (2007) and with our previous study [14,45,53,54,55,56].

The most important data pertain to the experiments conducted by pre-treating PMNs with DBV at 0.25 µg/mL before pooling with MRSA: the obtained killing percentages were significantly higher, with respect to controls, reaching values of about 56%. These results indirectly demonstrated that DBV penetrates into the neutrophils and boosts their intra-MRSA killing capability.

Notably, PMNs have a short lifespan and under homeostatic conditions undergo apoptosis to be removed by macrophages and avoid inflammation. Different works have evaluated the effect of antibiotics on neutrophil death. Healy D.P. et al. (2002) have demonstrated that clinical concentrations of eight different antibiotics, among which ceftazidime, gentamicin and ciprofloxacin were studied, had a modest but significant effect on the prolongation of survival [57]. On the other hand, the macrolides tilmicosin and tulathromycin enhanced neutrophil apoptosis [58,59]. Moreover, a recent work by Algorri and Beringer (2020) demonstrated that vancomycin, daptomycin and ceftaroline can increase, in a modest way, the survival of PMNs compared to unstimulated cells [60]. In the present research, we demonstrated that DBV up to plasmatic concentration had no effect on the survival of resting cells. No changes were also observed after stimulation with PMA—a potent cell activator shown to induce apoptosis, pyroptosis and NETosis [33]. The phagocytosis of MRSA by human PMNs leads to a particular form of necroptosis and lysis through activation of receptor interacting protein kinases-3 [61,62]. Here, we reported a significant increase in late apoptosis and necrosis of MRSA-activated PMNs. A delay in apoptosis was observed after pre-treatment with DBV indicating that the drug can, in part, subvert the rapid death of PMNs at the site of infection by reducing the risk of an exacerbated inflammatory response and tissue damage, and by favoring the killing of the bacteria.

Since ROS have been proposed to be correlated with bacterial killing, we explored ROS production in the DBV–neutrophils interaction. In *S. aureus*-activated cells, the ROS production was less pronounced than in PMA-activated cells; however, DBV pre-treatment only modestly increased ROS generation. The role of ROS in *S. aureus* killing is debated and the bacterium can produce different virulence factors that inhibit ROS mediated killing [21]. Moreover, in agreement with our data, glycopeptides did not modulate ROS, whereas for other antibiotics, contradictory results were reported [30]. Since we did not observe a significant change in ROS levels following DBV treatment, other mechanisms could be involved. Neutrophil antimicrobial proteins/peptide antimicrobial proteins such as myeloperoxidase, lactoferrin, metalloproteinases and a spectrum of neutrophil serine proteases contained in granules are critical for the effective functioning of neutrophils [36]. There have been reports of the synergy between antibiotics and antimicrobial peptides on the direct antimicrobial effect and on the enhancement of innate immune-mediated killing [63,64]. It is plausible that DBV-mediated intracellular killing of MRSA is potentiated by neutrophil antimicrobial peptides. In addition to direct immunomodulatory effects, DBV may also directly impact *S. aureus* virulence factor expression, which in turn may indirectly influence the activity of innate host defense peptides.

NETosis is an alternative mechanism of bacterial killing that can be modulated by antibiotics [65]. PMA is a potent *stimulus* often used to induce NETosis and degranulation in in vitro experiments [33]. Here, we reported no effect of DBV on the degranulation process induced by PMA, while an increase in NETosis was observed in PMA activated cells pre-treated with DBV. This agreed well with the increased production of ROS seen in PMA-induced neutrophils, as NETosis caused by PMA depends on NADPH oxidase. Other antibiotics were evaluated for the ability to modulate NETs release, and an inhibitory effect on PMNs was observed after treatment with gentamicin, azithromycin and chloramphenicol [65,66]; whereas, clindamycin can inhibit the nuclease activity of *S. aureus*, inducing NETs to eliminate the bacteria [67]. Other cell wall inhibitors, i.e., amoxicillin and fosfomycin, induced NETs release [49,68]; in particular, fosfomycin was found to enhance NETs in macrophages and neutrophils following *S. aureus* infection by inducing more entrapment and killing of the bacteria. These results are in line with ours, demonstrating a possible different effect on neutrophil functions, dependent on the antimicrobial mechanism of action. While NETs are important in innate immunity and resolution of infection, on the other end, an increase in NETs could represent a risk in stimulating autoimmune disorders. Therefore, an evaluation on the use of the antibiotics in some types of patients could be necessary to avoid detrimental consequences.

Neutrophils represent a source of pro-inflammatory cytokines and chemokines, which can act in paracrine and autocrine ways to mediate the inflammatory process. CXCL8 acts to recruit immune cells to the site of infection and has an important role in neovascularization, while TNFα is important to induce acute inflammation and is implicated in neutrophil survival/apoptosis [63,64]. Thus, in the present study, we evaluated the release of these cytokines in PMNs pre-treated with DBV at MIC and plasmatic concentration. The results highlighted an increase in the production of CXCL8 and TNFα in resting cells and in PMA and MRSA-activated cells at MIC value. The plasmatic concentration of DBV did not induce an increase in cytokine production, with respect to untreated PMNs. Moreover, a similar cytokine production was observed in MRSA stimulated cells, with respect to unstimulated ones. This effect is in line with the results obtained by Pichereau S. and colleagues (2012); they evaluated the effect of near and exceeding serum concentration of different antibiotics on peripheral blood mononuclear cells exposed to *S. aureus* and showed that the release of pro-inflammatory molecules was inhibited [67]. An immunomodulatory effect of the antibiotic, able to maintain a pro-inflammatory/anti-inflammatory cytokine balance, could be important to avoid detrimental consequences and improve the patient outcome in the case of severe infections with MRSA. Furthermore, it was reported that TNFα has the opposite effect on neutrophil survival with a high concentration able to induce apoptosis and a low concentration exerting apoptosis delay [68]. The survival of TNF-stimulated PMNs is CXCL8 dependent as demonstrated by Cowburn A.S. et al. (2004) [69]. As we here previously reported, a delayed apoptosis was observed in MRSA activated cells in the presence of DBV at plasmatic concentration, and this fit well with the results obtained in cytokine production.

## 4. Materials and Methods

### 4.1. Intracellular Killing Assay

A strain of MRSA clinically isolated, and frequently a cause of ABSSSIs [6], was plated on Mannitol Salt agar (Oxoid SpA, Rodano, Milan, Italy) and transferred into cryovials to be stored for an extended period at −80 °C [18].

Healthy donors were subjected to collection of peripheral venous blood into lithium heparin tubes (15 UI LH/mL blood) and maintained for 30 min at room temperature in a ratio 1:1 with 2.5% dextran (500,000 molecular weight; Sigma-Aldrich, Saint Louis, MO, USA) prepared in buffered saline solution. The supernatant containing leukocyte-rich plasma was gently layered on Ficoll-Paque (Pharmacia S.p.A., Milan, Italy) and centrifuged twice (1200 g, 15 min). A 30 s hypotonic shock in sterile distilled water was performed to lyse erythrocytes and to obtain pure PMNs that were counted in a Bürker chamber (Marenfield, Germany), resuspended in phosphate buffered saline (PBS; supplemented with 0.1% glucose and 0.1% human albumin) to reach 10^6^ cells/mL, and placed into sterile tubes containing RPMI 1640 added with 10% fetal calf serum (FCS). Before each experiment, the PMN viability, greater than 95%, was ascertained by trypan blue exclusion test [18,55,56].

The intracellular microbial killing exerted by PMNs against staphylococci (in a ratio of 10:1) in the presence of DBV (Sigma-Aldrich)—1xMIC, 0.25 µg/mL; 2xMIC, 0.5 µg/mL—was tested for 30, 60 and 90 min at 37 °C. Controls without DBV were also considered. The intraPMN killing values were measured, as punctually described in our previous works [18,19] and expressed as the survival index (SI), calculated by adding the surviving microorganism number at time zero to the survivor number at time X (T30, T60 and T90) and dividing by the survivor number at time zero. According to this formula, an SI = 1 corresponds to 100% microbial killing. Data were expressed as mean values ± SEM.

DBV-pre-treated PMN killing activity against MRSA and PMN killing activity against DBV-pre-treated MRSA were also tested. Briefly, PMNs or MRSA were independently incubated for 1 h with DBV (0.25 µg/mL or 0.5 µg/mL), and then bacteria or PMNs were added at 37 °C to permit phagocytosis for 30 min. The obtained PMN-bacterium mixtures were centrifuged (1200 g, for 5 min) and washed with phosphate saline to remove the free extracellular bacteria. After that, the intraPMN killing was evaluated as above mentioned [19].

At the end of the time killing, the culture supernatants of the pre-treated PMNs were stocked at −20 °C to determine the release of the pro-inflammatory cytokines (TNFα and CXCL8) analyzed by ELISA according to the manufacturer’s instructions (R&D Systems, Minneapolis, MN, USA) [18,19].

### 4.2. Phenotypic Analysis of PMN Apoptosis

Purified PMNs were harvested in polypropylene tubes at a concentration of 1 × 10^6^ in 0.5 mL RPMI 10% FCS without antibiotics and pre-treated for 60 min with DBV at MIC and at 25 µg/mL, a dose within the therapeutically achievable plasma concentration (20–30 mg/L) [70,71]. The cells were then stimulated with PMA (100 nM) or MRSA (10:1) for 3 h at 37 °C. After the incubation, cells were washed, resuspended in specific Buffer and stained with FITC-Annexin V and Propidium Iodide (PI) (Biolegend, San Diego, CA, USA), as indicated by the manufacturer’s instructions, to measure apoptosis. The analysis was performed using BD FACSCanto™ II Flow Cytometry System and the FlowLogic software v. 8.3 (Miltenyi Biotec, Bergisch Gladbach, Germany).

### 4.3. Oxidative Burst Assay

The ROS production was determined during the respiratory burst. Separated PMNs at 1 × 10^6^ in 0.5 mL RPMI 10% FCS without antibiotics were pre-treated in polypropylene tubes with different DBV concentrations (0.25 µg/mL and 25 µg/mL) for 60 min. Then, they were incubated with PMA (100nM), MRSA (10:1) or left untreated at 37 °C for 30 min with shaking. Subsequently, dihydrorhodamine 123 (DHR 123) solution (Sigma Aldrich), at a concentration of 30 μg/mL, was added to the stimulated and resting samples for 5 min. After centrifugation and two washes in PBS, the samples were resuspended in 1% paraformaldehyde to stabilize the cells and then analyzed by flow cytometry using BD FACSCanto™ II. A total of 10,000 events were collected for each sample and neutrophils were identified by their typical forward and sideward scatter characteristics [19].

### 4.4. Degranulation

Degranulation of separated PMNs was analyzed using BD FACSCanto™ II: 1 × 10^6^ PMNs were incubated with DBV (0.25 µg/mL or 25 µg/mL) for 60 min in 0.5 mL Hank’s Balanced Salt Solution (Thermo Fisher Scientific, Waltham, MA, USA) supplemented with 1% human serum albumin (Sigma Aldrich) and then treated with PMA (100 nM) for 30 min with shaking. The granularity degree of neutrophils was determined in the side scatter channel (SSC) after doublet discrimination and exclusion of dead cells using the LIVE/DEAD Fixable Violet stain (Thermo Fisher Scientific). Results are expressed as geometric mean fluorescence intensity (gMFI), which represents the cell density.

### 4.5. NET-Forming Capacity Assay

Purified neutrophils (1 × 10^6^) in 0.5 mL of RPMI base cell culture medium supplemented with calcium chloride (1 M) and 1% bovine serum albumin were incubated and pre-treated with 0.25 or 25 µg/mL of DBV for 60 min in a 24-well plate. Afterwards, the cells were stimulated with PMA (100 nM) for 4 h at 37 °C. After stimulation, neutrophils were washed to remove soluble neutrophil elastase that is not NET-associated. Next, 500 µL of nuclease (15 U/mL) was added to each well, incubated for 15 min at 37 °C and then inactivated by EDTA (500 mM). Supernatants were collected, centrifuged at 1200 rpm for 10 min and analyzed by NETosis Assay Kit as a release of neutrophil elastase (Cayman Chemical, Ann Arbor, MI, USA), as indicated by the manufacturer’s instructions.

### 4.6. Cytokine Release

The supernatants of PMNs pre-treated with DBV at MIC (0.25 µg/mL) or plasmatic concentrations (25 µg/mL) for 60 min and stimulated with PMA (100 nM) or MRSA (10:1) for 4 h, were collected to evaluate the cytokine release (CXCL8 and TNFα) by ELISA kit (R&D Systems, Minneapolis, MN, USA). The results are expressed as a percentage of release, with respect to untreated cells, considered as 100% [19].

## 5. Conclusions

*S. aureus* still represents a key pathogen in determining severe human infections, and the increasing isolation of drug-resistant strains—mainly MRSA—in the clinical setting poses a challenge in the choice of treatment management. PMNs are typically underappreciated professional phagocytes, regarding the history and advancements of contemporary immunology, but they play a pivotal role in *S. aureus* clearance and in determining the fate of infection. Unfortunately, *S. aureus* is able to evade immune response by several escaping strategies, in particular by inhibition of neutrophil recruitment, phagocytosis and staphylococcal killing.

Therefore, we investigated the potential immune modulating properties of DBV on PMNs towards MRSA. Our results suggested that DBV acts in synergism with neutrophils by enhancing staphylococcal killing. The pre-treatment with DBV did not affect respiratory burst and degranulation, while at the MIC concentration, it upregulated the release of CXCL8 and TNFα, and increased NETosis. In addition, DBV at human plasmatic concentration determined a delay in PMN apoptosis in the presence of MRSA. The DBV action on the boosted neutrophils’ capability to kill resistant *S. aureus* is not fully explainable by ROS and/or NET production, and suggests the involvement of oxidative independent mechanisms, or a DBV direct effect on the modulation of staphylococcal virulence factors.

Our observations highlighted that DBV should be considered as an outstanding therapeutic option for the management of serious *S. aureus* infections, sustained also by multi-drug resistant strains. Actually, it performs a double effect by acting both on MRSA and PMNs, and by reinforcing their clearance capability after the uptake of the pathogen.

## Figures and Tables

**Figure 1 ijms-24-02541-f001:**
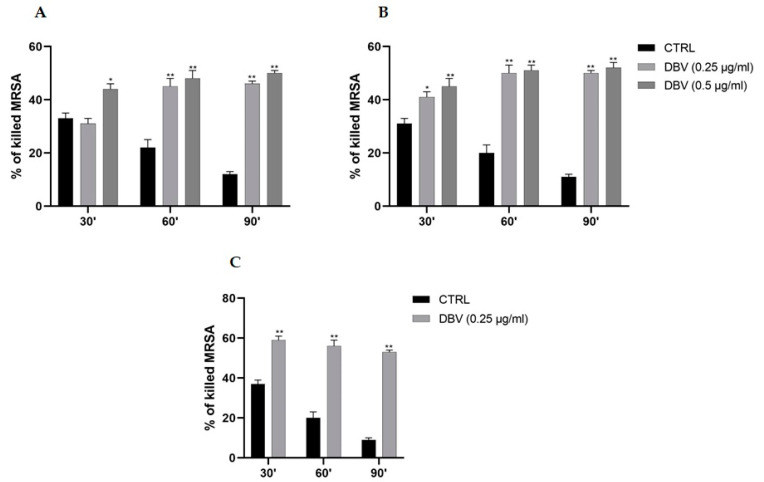
**DBV enhances intracellular MRSA killing activity of human PMNs.** PMNs (10^6^ cells/mL) and MRSA (10^7^ CFU/mL) were incubated with DBV (at MIC or 2xMIC) and their killing activity was evaluated at 30′, 60′ and 90′ (**A**). In pre-treated experiments, MRSA (**B**) or PMNs (**C**) were pre-treated for 60′ with DBV, then they were put together and the PMNs killing activity towards MRSA was evaluated. Data are shown as killing percentages from at least three independent experiments. * *p* < 0.05 ** *p* < 0.001 CTRL vs. DBV.

**Figure 2 ijms-24-02541-f002:**
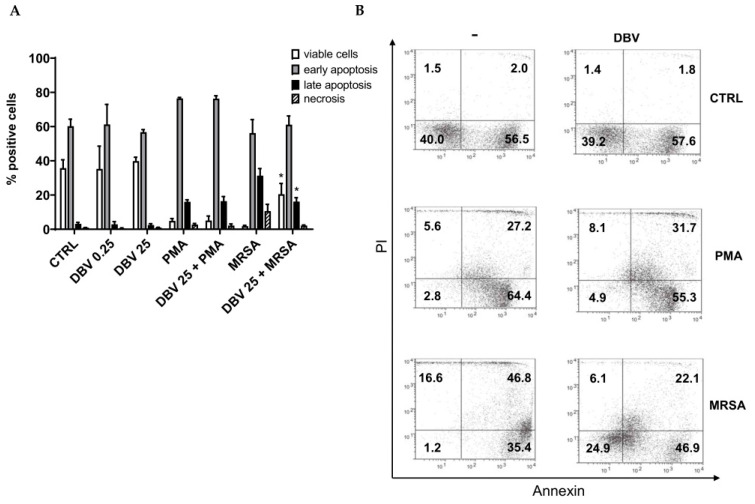
**DBV delays the apoptosis of MRSA activated-PMNs.** PMNs were untreated or treated with DBV at 0.25 and 25 μg/mL for 60′ and then unstimulated or stimulated with PMA (100 nM) or MRSA (10:1) for 3 h. Cell viability was evaluated by Annexin V FITC/PI staining. (**A**) Quantitative analysis of the viable, early apoptotic, late apoptotic and necrotic cells is shown. Values are means ± SEM of at least three independent experiments. * *p* < 0.05 vs. MRSA-treated PMNs. (**B**) A representative dot plot experiment of PMNs pre-treated with DBV at 25 μg/mL showing viable cells (Annexin neg/ PI neg cells, low left panel); early apoptotic cells (Annexin pos/PI neg cells, low right panel; late apoptotic cells (Annexin pos/PI pos cells, up right panel) or necrotic cells (Annexin neg/PI pos cells, up left panel) is shown. Values are means ± SEM of at least three independent experiments. * *p* < 0.05 vs. MRSA-treated PMNs.

**Figure 3 ijms-24-02541-f003:**
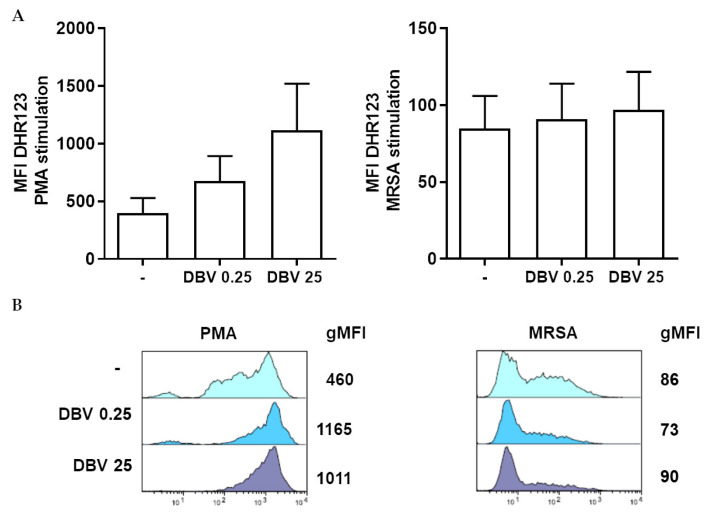
**Effect of DBV on ROS production by activated-PMNs.** (**A**) PMNs were untreated or treated with DBV at 0.25 and 25 μg/mL for 60′ and then stimulated with PMA (100 nM) or MRSA (10:1) for 30′. Cells were loaded with DHR 123, and the intracellular production of ROS was assessed by flow cytometry. The intensity of ROS production (measured as gMFI) is shown. Data show means ± SEM of at least three independent experiments. (**B**) Panels illustrating representative phenotypes displayed by PMNs exposed to PMA (left panel) or to MRSA (right panel), based on DHR 123 fluorescence expression. gMFI values are shown on the right side of each panel.

**Figure 4 ijms-24-02541-f004:**
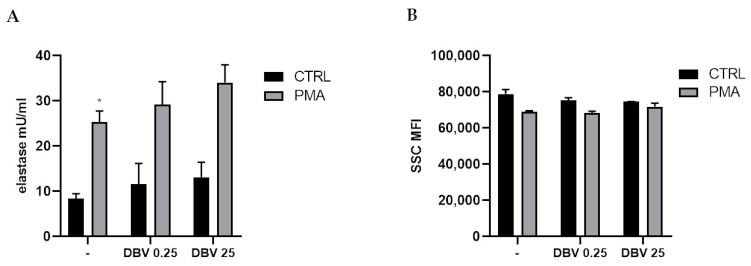
**Effect of DBV on NETosis and degranulation.** (**A**) To measure NETs production, PMNs were stimulated for 4 h with PMA (100 nM) after pretreatment with DBV (0.25 and 25 μg/mL). NETs were measured as NET-associated elastase. (**B**) PMNs were stimulated for 60′ with DBV (0.25 and 25 μg/mL) and then activated with PMA (100nM) for 30′. The degree of granularity was evaluated by FACS analysis in the side scatter channel (SSC) as MFI. Values are means ± SEM of at least three independent experiments. * *p* < 0.05 vs. CTRL.

**Figure 5 ijms-24-02541-f005:**
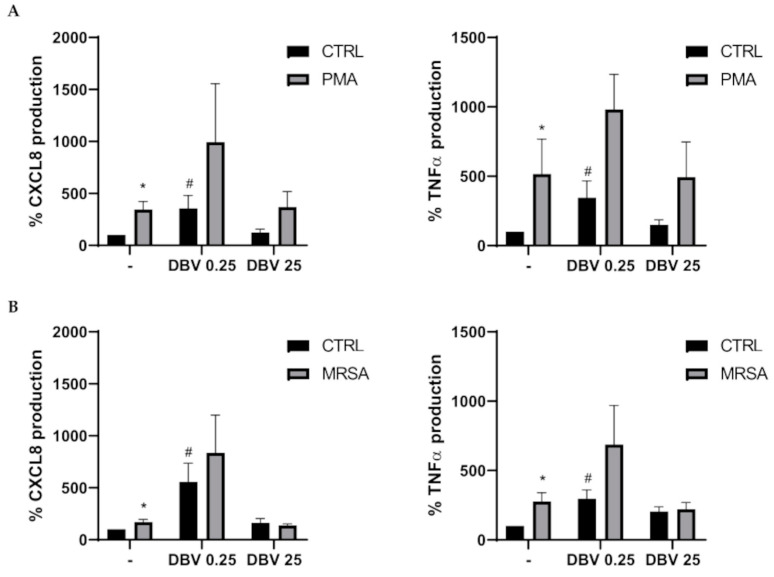
**Effect of DBV on cytokine release by PMNs.** PMNs were treated with DBV (0.25 and 25 μg/mL) and then activated with PMA (100 nM) (**A**) or MRSA (10:1) (**B**) for 4 h. The production of CXCL8 and TNFα in cell supernatants is shown with results expressed as percentage of release with respect to untreated cells (CTRL) considered as 100%. Values are means ± SEM of at least three independent experiments. * *p* < 0.05 vs. CTRL; ^#^
*p* < 0.05 vs. CTRL.

## Data Availability

The authors confirm that the data supporting the findings of this study are available within the article and/or on request from the corresponding author (V.A.).

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
