# Peer review of "Dalbavancin Boosts the Ability of Neutrophils to Fight Methicillin-Resistant Staphylococcus aureus"

_ijms, 2023, doi:10.3390/ijms24032541_

Round 1
Reviewer 1 Report
From the reviewer's point of view, this manuscript has flaws.
1. Authors should be more careful in interpreting the results of other researchers. In the Introduction section (lines 61-63) there is a controversial treatment of two literature reviews. Thus, the authors write "...erythromycin and azithromycin stimulate neutrophil's phagocytosis and intracellular killing of S. aureus [25].", although this review only considers how antibiotics interact with components of innate immunity, for example, with antimicrobial peptides; about erythromycin and azithromycin, which affect phagocytosis and killing, there is nothing there. Again, the authors write "...gentamicin and erythromycin can inhibit neutrophil functions [26]." If you read this review, you can find out that the effect of these antibiotics on both the function of neutrophils and their killing ability is varied, and, probably, one should not categorically assert about the exclusive inhibition of neutrophil functions by these antibiotics.
2. Materials and methods section.
It is not entirely clear why such antibiotic concentrations were chosen. For example, when measuring intracellular killing activity, 1 MIC and 2 MIC are used, 0.25 µg/ml and 0.5 µg/ml, respectively (line 347); when studying how the antibiotic affects the viability of neutrophils (line 368), the authors used 25 µg/ml, for oxidative burst (line 377) or degranulation (line 388), for some reason, two concentrations are already used (1 MIC, i.e. 0.25 µg/ml and 25 µg/ml). The reviewer believes that this choice of concentrations should be better explained.
3. Results section.
Subsection 2.1 describes how dalbavancin on neutrophil intracellular MRSA killing activity is affected. However, Figure 2 shows readers what proportion of bacteria die under certain conditions over a certain period. Is it correct to speak about killing activity here?
4. Discussion section. Obviously, the authors have their own point of view on this section.
Some of the material in this section is like the literature review in the introduction section. In addition, the authors for some reason avoid explaining the possibility of increasing the intracellular death of staphylococci due to the potentiation of the antibiotic by antimicrobial peptides of neutrophils. This is not very clear, since all the results presented in this manuscript show this, especially the lack of a noticeable effect on ROS production.
Reviewer 2 Report
The paper is interesting and nicely written.
The authors should improve English, for example:
MRSA is one of the most cause, change to: MRSA is one of the most common causes…
DBV determined a delay in the apoptosis of MRSA…, change to DBV caused delay…
Binding with, change to: „binding to“
Reviewer 3 Report
The article entitled "Dalbavancin boosts the ability of neutrophils to fight methicillin-resistant Staphylococcus aureus" presents an interesting topic about the antimicrobial impacts of neutrophils along with Dalbavancin against MRSA. MRSA causes one of the most troublesome conditions that are resistant to most antibacterial treatments. You used different concentrations of DBV against PMN-captured MRSA at various intervals to find out the best DBV concentration that has more lethality against MRSA. Pretreatment of bacteria/ PMN with DBV was an interesting part to consider any factors that could intervene in the assessment. In the end, DBV could be a supplemental treatment along with other therapies, however, more studies are needed to assert this claim. To improve the manuscript, please find some of my suggestions below:
- The manuscript needs to be polished grammatically. There are some grammatical mistakes. Some sentences are confusing and they need to be rewritten.
Line 11: contribute (remove s)
Line 13: causes
Line 20: use hyphen: MRSA-infected
Line 30: the bacterial cell wall
Line 40: add a coma after in particular
Line 49: contribute (without s)
Line 55: add hyphen: host-programmed
Line 57: and avoiding
Line 59: what are the other several immune cells? could you expand your explanation? please make more examples
Line 59: I suggest replacing this ''There has been increased interest, in recent years, in the influence of antibiotics on several immune cells including neutrophils"! with the following sentence: In recent years, more studies have been focused on the influence of antibiotics on the function of immune cells such as...
Line 61-63: Is it referred to the synergistic impacts of erythromycin-azithromycin and gentamicin-erythromycin? If yes, please indicate it in both sentences!
Line 68: to the killing of DBV
Line 81: slightly
Line 83: significantly
Line 92: with respect
Line 93: Remove "in detail"
Line 93-96: Please execute the following corrections:
Whereas for drug-free controls a decrease in PMN killing was observed when PMNs were in contact with DBV pre-treated MRSA (at 0.25 µg/ml), in which the killing values significantly (p<0.001) improved to 41% (SI=1.59), 50% (SI=1.50) and 50% (SI=1.50) after 30, 60 and 90 minutes of incubation.
Line 94: significantly
Line 98-102: Please implement the following changes: Similar results were obtained when PMNs were pre-treated with DBV before assessing its killing activity against MRSA, in which the killing values (p<0.001) saw a significant rise compared to controls reaching 59% (SI=1.41) after 30 minutes, 56% (SI=1.44) after 60 minutes, and 53% (SI=1.47) after 90 minutes of incubations (Figure 1C).
Line 103: Remove into..... penetrate PMNs
Line 115-118: Rewrite the sentence as follows: After MRSA exposure, the expression of the pro-inflammatory cytokines, specifically TNFα and CXCL8, was evaluated in the supernatants of DBV-pretreated PMNs, in which an augmented level of release related to both cytokines was recorded, even if it did not reach a statistically significant degree.
Line 119: statistically significant
Line 121: pretreated
Line 123: pretreated
Line 128: undergoing to a programmed apoptosis
Line 130: at a plasmatic concentration
Line 132: remove the hyphen in untreated
Line 134: Delayed apoptosis was observed
Line 140: delays the apoptosis
Line 145: necrotic
Line 146: is shown
Line 171: Parallel
Line 172: influence
Line 172: pretreated
Line 176: Another
Line 179: with respect....... PMA-treated
Line 183: measure (correct this word in other places as well)
Line 186: evaluated
Line 201: with respect
Line 344: Please add the abbreviation (FCS) in front of the fetal calf serum.
Line 368: Please define PMA wherever you mentioned it for the first time!
Line 376: Define FBS
And other similar cases...
Round 2
Reviewer 1 Report
The reviewer has no comments on this version of the manuscript.
Reviewer 2 Report
The authors have followed my suggestions.
Reviewer 3 Report
Thank you for implementing the corrections. I think that the manuscript has been much improved.